# Associations of lifetime concussion history and repetitive head impact exposure with resting-state functional connectivity in former collegiate American football players: An NCAA 15-year follow-up study

**Samuel R. Walton**[1,2,3,4]*, **Jacob R. Powell**[2,3], **Benjamin L. Brett**[5], **Weiyan Yin**[6], **Zachary Yukio Kerr**[1,2], **Mingxia Liu**[6], **Michael A. McCrea**[5], **Kevin M. Guskiewicz**[1,2], **Kelly S. Giovanello**[3,6,7]

1 Center for the Study of Retired Athletes, University of North Carolina at Chapel Hill, Chapel Hill, NC, United States of America, 2 Department of Exercise and Sport Science, University of North Carolina at Chapel Hill, Chapel Hill, NC, United States of America, 3 Cognitive Neuroscience of Memory Laboratory, University of North Carolina at Chapel Hill, Chapel Hill, NC, United States of America, 4 Physical Medicine and Rehabilitation, School of Medicine, Virginia Commonwealth University, Richmond, VA, United States of America, 5 Department of Neurosurgery, Medical College of Wisconsin, Milwaukee, WI, United States of America, 6 Department of Radiology, Biomedical Research Imaging Center, University of North Carolina School of Medicine, Chapel Hill, NC, United States of America, 7 Department of Psychology and Neuroscience, University of North Carolina at Chapel Hill, Chapel Hill, NC, United States of America

* samuel.walton@vcuhealth.org

**Data Availability Statement:** Data are available in an Open Science Framework repository, which

## Abstract

The objective of this study was to examine associations of lifetime concussion history (CHx) and an advanced metric of lifetime repetitive head impact exposure with resting-state functional connectivity (rsFC) across the whole-brain and among large-scale functional networks (Default Mode; Dorsal Attention; and Frontoparietal Control) in former collegiate football players. Individuals who completed at least one year of varsity collegiate football were eligible to participate in this observational cohort study ($n = 48$; aged 36–41 years; 79.2% white/Caucasian; 12.5±4.4 years of football played; all men). Individuals were excluded if they reported history/suspicion of psychotic disorder with active symptoms, contraindications to participation in study procedures (e.g., MRI safety concern), or inability to travel. Each participant provided concussion and football playing histories. Self-reported concussion history was analyzed in two different ways based on prior research: dichotomous "High" ($\geq 3$ concussions; $n = 28$) versus "Low" (<3 concussions; $n = 20$); and four ordinal categories (0–1 concussion [$n = 19$]; 2–4 concussions [$n = 8$]; 5–7 concussions [$n = 9$]; and $\geq 8$ concussions [$n = 12$]). The Head Impact Exposure Estimate (HIEE) was calculated from football playing history captured via structured interview. Resting-state fMRI and T1-weighted MRI were acquired and preprocessed using established pipelines. Next, rsFC was calculated using the Seitzman et al., (2020) 300-ROI functional atlas. Whole-brain, within-network, and between-network rsFC were calculated using all ROIs and network-specific ROIs, respectively. Effects of CHx and HIEE on rsFC values were examined using separate multivariable linear regression models, with a-priori $\alpha$ set to 0.05. We observed no statistically significant

may be found using the following DOI: 10.17605/ OSF.IO/B5ZN4.

**Funding:** KMG and MAM received funding for this study from the National Collegiate Athletic Association (https://www.ncaa.org/). The funders had no role in study design, data collection and analysis, decision to publish, or preparation of the manuscript.

**Competing interests:** I have read the journal's policy and the authors of this manuscript have the following competing interests: BLB reports grants and support from the National Institute of Neurological Disorders and Stroke and National Institute on Aging under the National Institutes of Health. KMG reports compensation from National Collegiate Athletic Association for other services and grants from Boston Children's Hospital (sub-award from the National Football League). MAM acknowledges researching funding from the National Institutes of Health, U.S. Department of Defense, Centers for Disease Control and Prevention, National Collegiate Association and National Football League (via subaward from Boston Children's Hospital). No other conflicts of interest are declared for the remaining authors.

associations between rsFC outcomes and either CHx or HIEE ($ps \geq .12$). Neither CHx nor HIEE were associated with neural signatures that have been observed in studies of typical and pathological aging. While CHx and repetitive head impacts have been associated with changes in brain health in older former athletes, our preliminary results suggest that associations with rsFC may not be present in early midlife former football players.

## Introduction

Several factors increase the risk for clinically significant cognitive decline in aging individuals (e.g., mild cognitive impairment [MCI] and dementia) [1–3]. Among the most reported risk factors for collision-sport athletes are exposures to repetitive head impacts and traumatic brain injuries (TBIs) [4–8]. Sport-related concussions (SRCs), a form of mild TBI [9], occur commonly in American football players [10, 11]. Studies have suggested that former football players may have earlier onset of, and/or increased risk for, cognitive decline or neurodegenerative disease diagnoses compared with the general population, purportedly due to their exposure to repetitive head impacts—SRC or otherwise—during football play [4–6, 8]. It is vital, therefore, to develop the scientific understanding of what constitutes normal versus abnormal changes in brain function as these players age. To do so requires measuring the associations between pathological changes and pertinent risk factors for neurodegenerative changes such as repetitive head impact exposure and traumatic brain injuries (e.g., SRC) sustained throughout a sporting career.

One method for examining biologic changes in brain health is resting-state functional activity as measured with magnetic resonance imaging (MRI). Changes in functional connectivity (correlated patterns of neural activity between brain regions) within large-scale functional brain networks, such as the Default Mode (DMN) [12, 13], Dorsal Attention (DAN) [14], and Frontoparietal Control (FPCN) [15] networks, have been characterized among aging individuals with age-related cognitive changes as well as those with clinically significant cognitive declines (e.g., those with MCI and dementia-related disorders) [16–22]. Specifically, overall resting-state functional connectivity (rsFC) may be decreased between regions within the DMN and DAN and may be stronger between the FPCN and both the DMN and DAN in cognitively normal older individuals (e.g., 60 years of age and over) compared with younger adults [17–20]. Hyperconnectivity between networks may represent an altered ability of the FPCN to serve as a "circuit breaker" [15] for activity within the DMN and DAN, which may partially explain age-related cognitive and mood-related behavior declines [17, 20]. The similarity between these characteristic network-based rsFC changes in those who experience age-related cognitive changes and what has been reported in studies of individuals with clinically significant cognitive declines (e.g., MCI and dementia) theoretically suggests that individuals with clinically significant cognitive declines may be experiencing exacerbated or accelerated biological aging processes [23].

Short-term and long-term changes to rsFC have been described in active adolescent and young adult athletes following SRC, and these changes largely resemble the patterns of functional reorganization (i.e., changes to within- and between-network rsFC in large-scale networks) observed in individuals experiencing typical aging and neurodegenerative disease processes [7, 18, 19, 21, 22, 24, 25]. There is mixed evidence as to whether rsFC changes persist beyond clinical recovery from SRC [7, 24, 26], and studies have reported neural recruitment differences when performing a cognitive task in former football players aged between 50–75

years with three or more lifetime SRCs as compared to those with fewer, despite no observed differences in task performance [27, 28]. Further evidence suggests that rsFC changes also occur in relation to repetitive head impacts, even without overt clinical signs or symptomsthat are consistent with a concussion diagnosis [29]. Taken together, long-term alterations in functional connectivity may result from greater exposure to SRC and repetitive head impacts, even in relatively young individuals, and these changes may resemble those observed in individuals experiencing cognitive decline. In this light, it is also reasonable to consider functional connectivity changes as potential biomarkers for the advanced aging and the early onset of pathological changes in former football players. In this preliminary study, we measured rsFC across the whole brain, and among large-scale functional brain networks (DMN, DAN, and FPCN) in a sample of early midlife former collegiate football players to examine whether rsFC was associated with lifetime concussion history or an advanced metric of lifetime repetitive head impact exposure. We hypothesized that alterations in rsFC as the result of head impacts sustained while playing football, if present, would manifest with network connectivity patterns similar to those described above in healthy older adults, including: 1) lower whole-brain rsFC; 2) lower within-network rsFC in the DMN and DAN; and, 3) higher between-network connectivity between the FPCN and both the DMN and DAN.

## Materials & methods

### Participants

Participants included former collegiate football players who completed an online general health survey in 2014 as part of a larger study of former collegiate athletes approximately 15-years after completion of their collegiate sport participation (the *National Collegiate Athletic Association [NCAA] 15-Year Follow-up Study*) [30]. The general health survey used in this study was adapted from previous studies of former football players with input from epidemiologists, athletic trainers, neuropsychologists, physicians, and former football players [4, 30]. Former collegiate football players who completed the survey were recruited to participate in a comprehensive in-person evaluation (e.g., neuroimaging, neurocognitive testing, and patient-reported outcome measures). Each player was contacted by a research coordinator, and those who responded were screened for eligibility. Inclusion criteria for the in-person visit was participation in at least one year of collegiate football. Exclusion criteria were a history of psychotic disorder with active symptoms, any contraindications to participation in study procedures (e.g., MRI safety concern), or inability to travel. This study was approved by the Institutional Review Boards at the University of North Carolina at Chapel Hill and the Medical College of Wisconsin, and all participants provided written informed consent prior to participation.

### Image acquisition

Participants completed study visits at one of two separate institutions, and all images were acquired via 3T magnetic resonance scanner with a 32-channel head coil (Siemens MAGNETOM Prisma at the University of North Carolina at Chapel Hill; GE Healthcare Signa Premier at the Medical College of Wisconsin). During the course of data collection, there was a scanner software update at the Medical College of Wisconsin, resulting in three closely-related sets of acquisition parameters for each of the structural and functional MRI series (Table 1). T1-weighted Magnetic Prepared Rapid Gradient Echo (MPRAGE) and resting-state functional MRI (rsfMRI) blood oxygen level-dependent (BOLD) scans were collected for each participant.

**Table 1.  Magnetic resonance imaging (MRI) acquisition parameters across study sites.**

| Label | University of North Carolina at Chapel Hill Siemens Magnetom Prisma | | Medical College of Wisconsin GE Premier (pre-software update) | | Medical College of Wisconsin GE Premier (post-software update) | |
|---|---|---|---|---|---|---|
| | *n* = 38 | | *n* = 4 | | *n* = 6 | |
| | T1w | rsfMRI | T1w | rs-fMRI | T1w | rsfMRI |
| *Acquisition time* | 4m:54s | 7m:00s | 4m:07s | 6m:51s | 5m:21s | 6m:50s |
| *Plane* | Sagittal | Axial | Sagittal | Sagittal | Sagittal | Axial |
| *Slices* | 176 | 72 | 160 | 72 | 184 | 72 |
| *Matrix* | 256x208 | 104x104 | 256x204 | 104x104 | 256x256 | 104x104 |
| *TE (ms)* | 2.03 | 33 | 7.592 | 33.1 | 2.016 | 21.8 |
| *TR (ms)* | 2540 | 802 | 3.008 | 802 | 4.672 | 800 |
| *FOV (cm)* | 256x208 | 210x210 | 256x256 | 210x210 | 256x256 | 208x208 |
| *Thickness (mm)* | 1.0 | 2.0 | 1.0 | 2.0 | 1.0 | 1.5 |
| *Volumes* | 1 | 512 | 1 | 512 | 1 | 512 |

All series were acquired using a 32-channel head coil and a 3T magnet at each study site. T1w = T1-weighted images (3D Magnetization Prepared Rapid Gradient Echo [MPRAGE]); rsfMRI = resting-state functional MRI (Blood-Oxygen-Level-Dependent [BOLD] signal).

## Data processing

Structural and functional MRI images were processed following a typically used pipeline [31, 32]. Specifically, T1 MPRAGE images were pre-processed for each participant using Freesurfer. The brain structural images were segmented into grey matter, white matter, and cerebrospinal fluid (CSF), and each resultant image was visually inspected by two of the study team members (SRW & JRP) for reconstruction errors. Functional data (rsfMRI) were preprocessed using FSL [33–35]. The preprocessing steps included discarding the first 10 volumes for magnetization equilibrium before processing, motion correction, and bandpass filtering (0.01–0.08 Hz). Mean signal from white matter, CSF, whole brain, and 24 motion parameters were removed using a linear regression model. In order to further reduce the motion effect, FD-DVARS "scrubbing" approach was applied [36]. Subsequently, rsfMRI images were aligned to the corresponding T1-weighted images by using linear alignment. And the alignment between T1-weighed images and Montreal Neurological Institute (MNI) template was performed by using the advanced normalization tools (ANTs) [37]. To improve the accuracy of registration, brain tissue segmentation images were employed to calculate the deformation field to the MNI template, as well as the reverse deformation field from MNI template to each individual subject.

Using the brain atlas provided by Seitzman et al. [38], deformation back to the individual space was performed to extract the mean time-series BOLD signal for each of 300 regions of interest (ROIs). This atlas was selected as it contains multiple ROIs in subcortical grey matter structures and pre-defines large-scale resting-state networks by uniquely assigning specific ROIs to a single network (or otherwise "unassigned" designation). Pearson's correlation coefficients (*r*) were calculated between all pairs of ROIs for each subject.

## Outcome measures (rsFC values)

Whole-brain rsFC was calculated as the average Pearson *r* correlation value across the BOLD signal time-series between each of the 300 individual functional ROIs. To test our hypotheses, within-network and between-network average Pearson *r* correlation values were calculated using the pre-defined DMN (65 ROIs), DAN (16 ROIs), and FPCN (36 ROIs) network nodes [38]. Within-network average rsFC was calculated as the average between ROI Pearson *r*

correlation value between all pairs of ROIs dwelling within a given network (e.g., DMN). Between-network average rsFC was calculated as the average between ROI Pearson *r* correlation value between each individual node of one specific network and each individual node of another specific network. These operationalizations resulted in seven dependent variables: 1) whole-brain average rsFC; 2) within-DMN average rsFC; 3) within-DAN average rsFC; 4) within-FPCN average rsFC; 5) DMN-DAN average rsFC; 6) DMN-FPCN average rsFC; and 7) DAN-FPCN average rsFC.

## Concussion history & head impact exposure estimate

History of concussion was self-reported by each participant using a definition that has been employed in previous research with current and former athletes [30, 39]. This operational definition described concussion as, "an injury occurring typically, but not necessarily, from a blow to the head, followed by a variety of symptoms that may include any of the following: headache, dizziness, loss of balance, blurred vision, 'seeing stars,' feeling in a fog or slowed down, memory problems, poor concentration, nausea, throwing up, and loss of consciousness" [30, 39]. This method of self-reporting concussion history has shown moderate levels of consistency (weighted Cohen κ = 0.48) over repeated administrations separated by many years [40]. After reading the operational definition of a concussion, participants then reported the total number of lifetime concussions they had sustained through sport or other mechanisms (e.g., military service, motor vehicle accidents).

Lifetime exposure to repetitive head impacts without diagnosed injury (e.g., head impacts that did not result in overt clinical signs or symptoms) were estimated using the adjusted Head Impact Exposure Estimate (HIEE) [41, 42]. A 30-minute structured interview was used to gather information regarding participation in contact football games and practices across each individual year of football participation at the high school, collegiate, and professional levels [41, 42]. For each year of play, participants detailed their primary and secondary playing positions, the average number and length (hours) of practices during each week of the pre-, regular, and post-season participation, the number of games played, and an estimate of the percentage of time playing in each game that year (0%; 25%; 50%; 75%; or 100%). These data were used to calculate a "number of contact hours" estimate for each participant. Those estimates were then adjusted to account for the number of head contacts that might be sustained for each individual by player position and level of play based on previous reports that utilized helmet-mounted accelerometers to measure head impacts [41, 43, 44]. The resultant number (HIEE) serves as a surrogate for the estimated number of head impacts to which that individual had been exposed during football participation in high school and beyond.

## Analyses

Primary independent variables were lifetime self-reported concussion history and HIEE. Concussion history was operationalized in two separate ways based on common standards in the existing literature: A) dichotomous "High" ($\geq$ 3 concussions; *n* = 28) versus "Low" (< 3 concussions; *n* = 20) history groups; and B) four ordinal categories (0 or 1 concussion [*n* = 19]; 2 to 4 concussions [*n* = 8]; 5 to 7 concussions [*n* = 9]; and 8 or more concussions [*n* = 12]). The dichotomous operationalization was selected to be similar to prior research examining the long-term effects of concussions on fMRI outcomes and cognitive function, where outcomes from participants three or more lifetime concussions were contrasted with those reporting fewer [4, 27, 28]. Further, ordinal operationalization of concussion history in recent studies has allowed for a more granular investigation of the effects of multiple concussions on long-term brain health [8, 45, 46], and we opted to explore (i.e., as a sensitivity analysis) whether the

same phenomenon would be observed in the present study by ascribing previously used concussion history groupings from overlapping study samples [42, 45]. The HIEE measure was included in each model as a continuous variable, regardless of the concussion history operationalization.

Potential covariates were participant age, body mass index (BMI), and MRI acquisition site. Analyses were performed to test univariable associations between each potential covariate and all seven of the rsFC outcomes of interest (S1 File). Both BMI and MRI acquisition site were related to one or more of the outcomes individually, and were therefore included as covariates in the multivariable models.

Separate multivariable linear regression models were fit for each of the seven rsFC outcomes including our primary exposures of interest (concussion history and HIEE) as well as covariates (BMI, and MRI acquisition site) as independent variables. Altogether, there were a total of 14 models fit (seven for each operationalization of lifetime concussion history). As this was a preliminary study, we set a-priori $\alpha$ at 0.05, and we've interpreted results based on these values as well as measures of effect size (standardized beta values [β]). All analyses were performed with SPSS version 28.0 (Armonk, NY). Post hoc observed power for each regression was calculated in G*Power v3.1.9.7.

## Results

### Participants

Participants were recruited from a sample of former collegiate athletes who previously completed a general health survey [30]. Initially, 123 former collegiate football players were able to be reached for in-person visit screening. Among these former players, 65 opted not to participate, met exclusion criteria (e.g., for MRI safety reasons), or did not respond to the study team. As a result, 58 former players completed in-person visits. Generally speaking, the cohort fell within the average range across indices of neurobehavioral function as described previously by Brett et al. [42]. Three of these participants did not participate in the MRI portion of the study due to claustrophobia (n = 2) or an acquisition protocol deviation (n = 1). A total of 55 male former collegiate football players participated in the MRI study. Among these participants, 7 (12.7%) had missing (n = 1) or unusable rsfMRI data (n = 6) due to poor functional image resolution precluding measurement of BOLD signal in one or more ROI. The resulting sample size was 48 participants across both research sites (Table 2).

### Resting state functional connectivity

After adjusting for BMI and MRI acquisition site, there were no statistically significant differences ($ps \geq 0.30$) for any of the rsFC outcomes between those with "High" (three or more) versus "Low" (less than three) lifetime concussion history (Table 3; Figs 1 and 2). Similarly, when concussion history was operationalized into 4 ordinal categories, no statistically significant associations were observed ($ps \geq 0.12$) between concussion history and any of the rsFC outcomes (Table 3; Figs 3 and 4). Finally, HIEE was not significantly associated with rsFC outcomes in any of the multivariable models ($ps \geq 0.14$). We also computed standardized effect sizes for both concussion history and HIEE (i.e., β-values). The largest β-values were observed for HIEE in models for within-FPCN rsFC and between-network rsFC for DMN-DAN (Table 3). Specifically, larger HIEE was associated with lower within-FPCN and higher DMN-DAN rsFC values, regardless of the operationalization variable used for concussion history. Plots of bivariate associations between HIEE and each of the seven rsFC outcomes are in Fig 5. Observed power estimates for each model ranged from 25–91% and are presented in S1 File.

**Table 2. Participant characteristics.**

|  | Full sample | "Low" lifetime concussion history group (<3 concussions) | "High" lifetime concussion history group (≥3 concussions) |
|---|---|---|---|
|  | *n* = 48 | *n* = 20 | *n* = 28 |
| **Age**[a], mean (standard deviation) | 37.9 (1.5) years | 37.9 (1.3) years | 37.9 (1.6) years |
| **Body Mass Index (BMI)**, mean (standard deviation) | 30.6 (4.3) kg·(m$^2$)$^{-1}$ | 31.1 (4.6) kg·(m$^2$)$^{-1}$ | 30.2 (4.1) kg·(m$^2$)$^{-1}$ |
| **Race**, n (%) |  |  |  |
| White or Caucasian | 38 (79.2) | 14 (70) | 24 (85.7) |
| Black or African American | 7 (14.6) | 3 (15) | 4 (14.3) |
| Multiracial | 3 (6.3) | 3 (15) | 0 (0) |
| **MRI acquisition site**, n (%) |  |  |  |
| University of North Carolina at Chapel Hill | 38 (79.2) | 16 (80.0) | 22 (78.6) |
| Medical College of Wisconsin | 10 (20.8) | 4 (20.0) | 6 (21.4) |
| **Lifetime concussion history**, median (lowest, highest) | 4 (0,24) | 1 (0,2) | 6.5 (3,24) |
| **Played professional football after college**, n (%) | 7 (14.6) | 2 (10) | 5 (17.9) |
| **Total years of football play**, mean (standard deviation) | 12.5 (4.4) | 13.2 (3.2) | 11.9 (5.0) |
| **Adjusted Head Impact Exposure Estimate (HIEE)**, mean (standard deviation) | 1292.0 (448.3) | 1262.4 (504.7) | 1313.0 (411.8) |

All participants were male.

[a] Ages ranged from 36 to 41 years old.

## Discussion

Our findings suggest that lifetime concussion history and accumulated head impact exposure among younger former collegiate football players were not significantly associated with functional connectivity of large-scale brain networks associated with aging during early midlife. However, there were notable effect sizes suggesting a relationship between repetitive head impacts (HIEE) and functional connectivity profiles that have been associated with the aging process (lower within-network connectivity and higher between-network connectivity). Previously reported findings related to age-related cognitive decline, MCI, and dementia in older former football players suggest that these players may be at increased risk of developing dementia-related disorders or accelerated cognitive aging [4, 5, 8]; however evidence for causal relationships between concussion history, repetitive head impacts, and these clinical outcomes have not been established. The present study provides evidence that broad changes to neural activity in the brain in relation to accumulated head trauma from football participation may not be readily detectable (i.e., too subtle to measure) or absent in individuals approximately 15-years after their collegiate sport participation. Continued follow-up with these participants and further evidence from prospective, longitudinal monitoring of brain health in former athletes is imperative to develop an understanding of change over time and associations of long-term brain health with head trauma.

In addition to our primary categorization of concussion history into "High" (three or more) and "Low" (less than three) lifetime injury groups, we sought to explore whether a more granular, ordinal categorization scheme might identify patterns with increasing exposure to injury that could be hidden in the traditional dichotomous operationalization, as has been seen in recent research [8, 45, 46]. Neither of the operationalizations of concussion history (dichotomous or ordinal categories) used in our study were associated with intra- or inter-network rsFC of the DMN, DAN, and FPCN networks or whole-brain rsFC. There are limitations inherent to retrospective recall of lifetime concussion history that warrant consideration when

**Table 3. Standardized β-values from multivariable linear regression models.**

| | "High" vs. "Low" Concussion History | | 4-Category Concussion History | |
|---|---|---|---|---|
| | Self-Reported Concussion History | | | |
| Outcome | Standardized β | *p*-value | Standardized β | *p*-value |
| **Whole-brain** | .115 | .41 | .061 | .66 |
| **Within-network** | | | | |
| DMN | .010 | .95 | .089 | .54 |
| DAN | .021 | .88 | .137 | .33 |
| FPCN | .071 | .62 | -.024 | .87 |
| **Between-network** | | | | |
| DMN-DAN | .001 | .99 | -.169 | .24 |
| DMN-FPCN | .152 | .30 | .009 | .95 |
| DAN-FPCN | .054 | .70 | .122 | .37 |
| | Adjusted Head Impact Exposure Estimate | | | |
| Outcome | Standardized β | *p*-value | Standardized β | *p*-value |
| **Whole-brain** | -.045 | .75 | -.037 | .80 |
| **Within-network** | | | | |
| DMN | .011 | .94 | .008 | .96 |
| DAN | -.159 | .29 | -.163 | .27 |
| FPCN | -.231 | .12 | -.224 | .14 |
| **Between-network** | | | | |
| DMN-DAN | .181 | .24 | .188 | .21 |
| DMN-FPCN | -.041 | .72 | .067 | .66 |
| DAN-FPCN | .027 | .85 | .026 | .85 |

Multivariable models included self-reported concussion history and adjusted Head Impact Exposure Estimates (HIEE) as predictors for each resting-state functional connectivity outcome. Two sets of models were fit with concussion history operationalized as either binary ("High" vs. "Low") or the 4-category operationalization (0 or 1 concussion; 2 to 4 concussions; 5 to 7 concussions; 8 or more concussions) based on previous studies with these participants. Both body mass index (BMI) and acquisition site were observed to have significant univariable associations with one or more of the outcome variables of interest and were included in each multivariable linear regression model as covariates. Adjusted $R^2$ values for multivariable linear regression models with concussion history operationalized as "High" vs. "Low" were: whole-brain (.120); within-DMN (.000); within-DAN (.066); within-FPCN (.078); DMN-DAN (.012); DMN-FPCN (.028); and DAN-FPCN (.137). Adjusted $R^2$ values for multivariable linear regression models with concussion history operationalized as 4 categories were: whole-brain (.110); within-DMN (.009); within-DAN (.087); within-FPCN (.073); DMN-DAN (.043); DMN-FPCN (.003); and DAN-FPCN (.150).

interpreting these findings, and previous work has described moderate consistency in recall over time [40]. Despite this, self-reported concussion history has previously been associated with increased rsFC within the DMN in current collegiate athletes [7], altered neural recruitment patterns in former football players aged between 50–75 years when completing a cognitive task [27, 28], and with self-reported cognitive dysfunction and atypical cognitive decline [4, 8, 46]. However, there is little research with former football players under 50 years of age related to the long-term effects of concussion history and repetitive head impacts on brain health.

Our recent work with an overlapping sample of relatively young (36–41 years of age) former football players noted significant associations between greater lifetime HIEE and multiple aspects of neurobehavioral functioning–including worse subjectively reported cognitive function, general psychological distress, and executive functioning alongside worse objectively measured memory and processing speed task performances [42]. In that study, we also observed that concussion history did not significantly alter the associations between neurobehavioral functioning and HIEE, despite these two measures of exposure to head trauma being

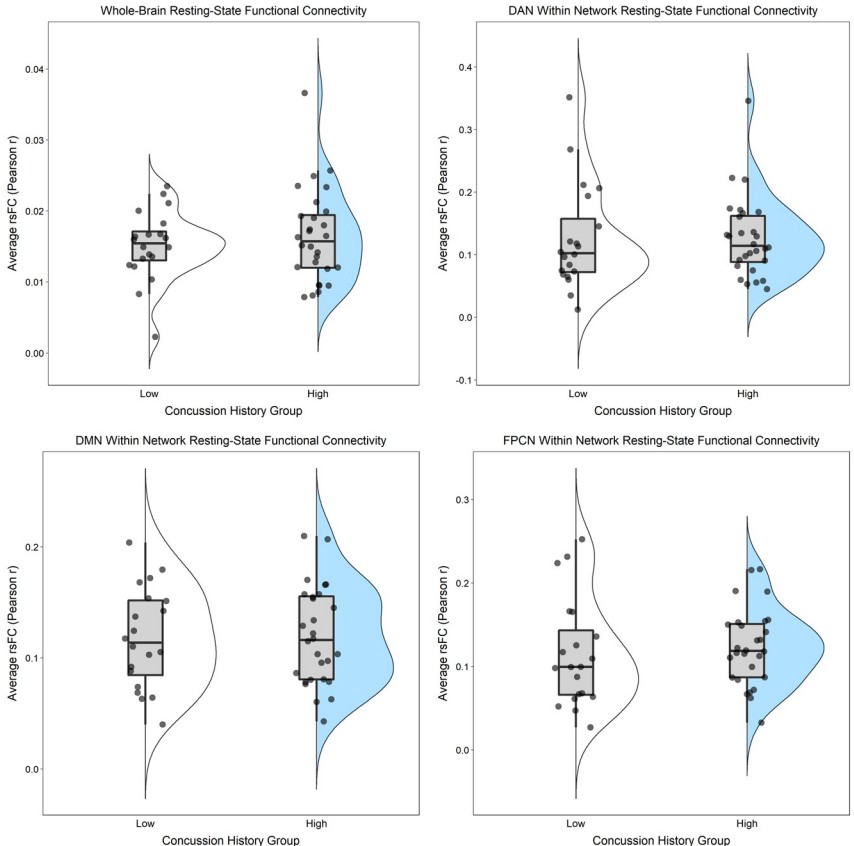

**Fig 1. Whole-brain and within-network resting-state functional connectivity (rsFC) in large-scale networks separated by dichotomous concussion history group.** Whole-brain rsFC (top-left) was calculated as the average Pearson *r* correlation (y-axes) between each of the 300 regions of interest (ROIs) and all other ROIs across the time-series. Within-network rsFC for the Dorsal Attention Network (DAN; top-right), the Default Mode Network (DMN; bottom-left), and the Frontoparietal Control Network (FPCN; bottom-right) was calculated as the average Pearson *r* correlation between each of the individual ROIs from the given network and all other ROIs within that same network across the time-series. Individual points represent participant-level outcomes for each measure within each of the concussion history groups (White = "Low" [fewer than 3 lifetime concussions]; Blue ="High" [3 or more lifetime concussions]). Boxplots represent the median and interquartile range, and the whiskers extend to 1.5 times the interquartile range. The violin plot is a depiction of the density of individual rsFC values for each measure.

distinct from one another [42]. The lack of statistically significant associations between HIEE and rsFC outcomes in the present study suggests that alterations in rsFC within and between large-scale networks may not be the underlying processes (i.e., neural correlates) for the relationship between HIEE and neurobehavioral functioning, or they may require more precise measurement to detect. It is possible that the association between exposure to head trauma and rsFC may be observed by using other approaches that describe functional organization in the brain (e.g., graph theoretical measures like small world topology) [47, 48]. These approaches warrant investigation in former football players as they may provide insight into latent constructs of brain health such as communication efficiency and resilience [49, 50], and some graph measures have even been associated with early recognition of neurodegenerative changes [51, 52].

The functional connectivity data reported in this study are the first to be reported in relation to concussion and repetitive head impact history in participants of this age. Specifically, objective markers of biological brain health in former athletes below 50 years of age are

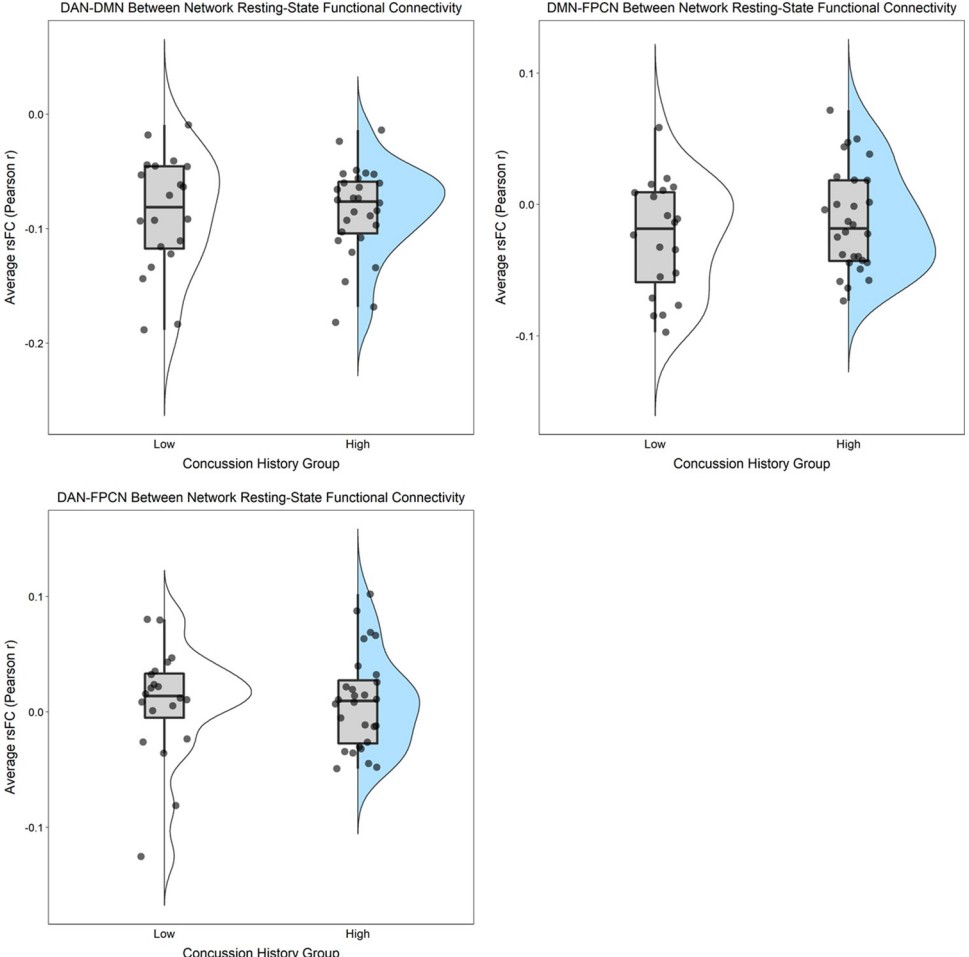

**Fig 2. Between-network resting-state functional connectivity (rsFC) in large-scale networks separated by dichotomous concussion history group.** Between-network rsFC was calculated as the average Pearson *r* correlation (y-axes) between each of the individual ROIs from one specific network and all individual ROIs in another specific network across the time-series: Dorsal Attention Network and Default Mode Network (top-left); Default Mode Network and Frontoparietal Control Network (top-right); and Dorsal Attention Network and Frontoparietal Control Network (bottom). Individual points represent participant-level outcomes for each measure within each of the concussion history groups (White = "Low" [fewer than 3 lifetime concussions]; Blue ="High" [3 or more lifetime concussions]). Boxplots represent the median and interquartile range, and the whiskers extend to 1.5 times the interquartile range. The violin plot is a depiction of the density of individual rsFC values for each measure.

relatively understudied compared to their older counterparts. It is notable that only a few (n = 7) of the participants in this study played football after their collegiate careers while also reporting 12.5 years of football play, on average. This sample is therefore mostly representative of amateur athletes who began playing football at youth levels. Research with former athletes in early midlife—especially longitudinal studies—are essential to understanding the relationships between head trauma and the aging process across the lifespan. Participants in this study with more self-reported lifetime concussions and greater HIEE did not exhibit functional connectivity differences when compared to those with fewer concussions and/or lower HIEE; however, these participants are purportedly at higher risk of developing early cognitive decline and neuropathology associated with head trauma according to previous research findings [4, 5, 8]. Therefore, this timepoint is a key contribution to the literature in that longitudinal

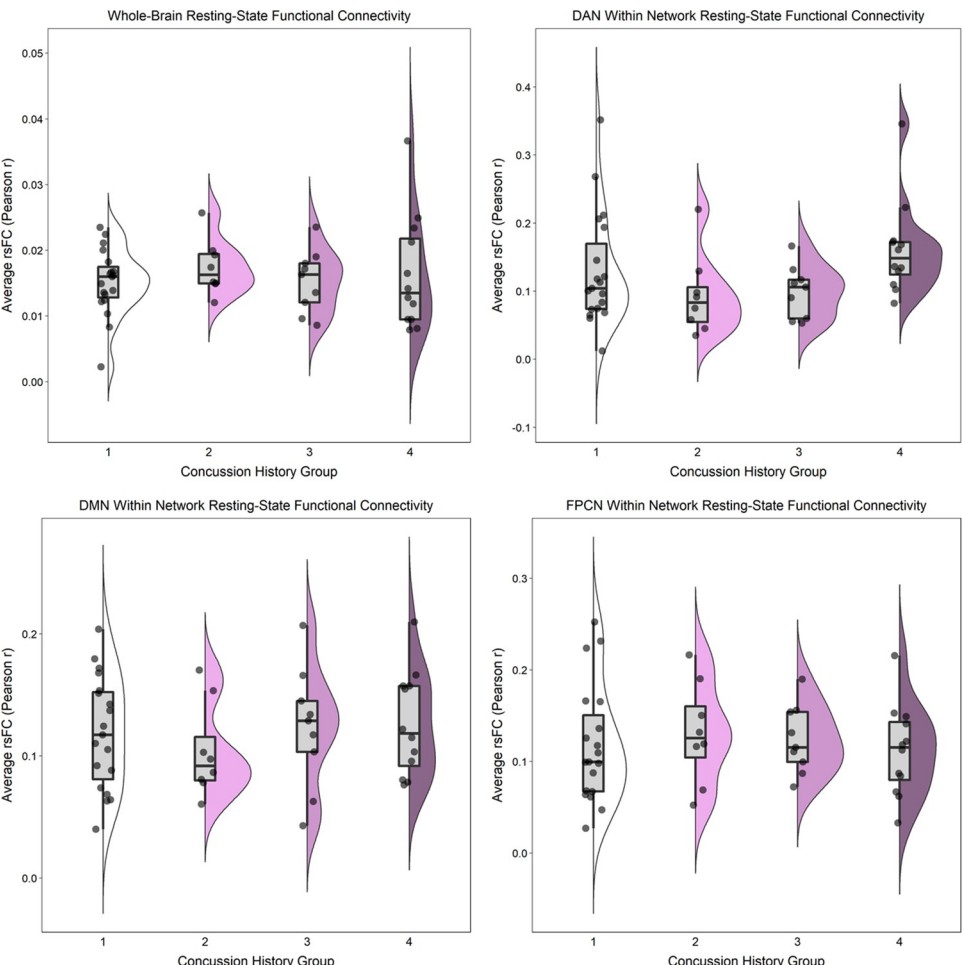

**Fig 3. Whole-brain and within-network resting-state functional connectivity (rsFC) in large-scale networks separated by four ordinal concussion history groups.** Whole-brain rsFC (top-left) was calculated as the average Pearson *r* correlation (y-axes) between each of the 300 regions of interest (ROIs) and all other ROIs across the time-series. Within-network rsFC for the Dorsal Attention Network (DAN; top-right), the Default Mode Network (DMN; bottom-left), and the Frontoparietal Control Network (FPCN; bottom-right) was calculated as the average Pearson *r* correlation between each of the individual ROIs from the given network and all other ROIs within that same network across the time-series. Individual points represent participant-level outcomes for each measure within each of the concussion history groups (1 = 0 or 1 concussion [*n* = 19]; 2 = 2 to 4 concussions [*n* = 8]; 3 = 5 to 7 concussions [*n* = 9]; and 4 = 8 or more concussions [*n* = 12]). Boxplots represent the median and interquartile range, and the whiskers extend to 1.5 times the interquartile range. The violin plot is a depiction of the density of individual rsFC values for each measure.

follow-up will allow us to investigate the interaction between head trauma and age, especially over the next decade as these participants approach 50 years of age.

One limitation of this study is the potential for error in data relying on self-report. Specifically, participants self-reported the primary exposure measures used in this study (concussion history & HIEE), and it is possible that self-reported exposure differs from the true incidence of exposure. Further, data were collected at two research sites and under three separate image acquisition protocols. To address this acquisition heterogeneity, we evaluated the association between study site and the rsFC outcomes (S1 File), and ultimately controlled for site in the analyses. Data from the present study may also not be representative of former collegiate football players at large, and should be considered in this light. Namely, there were only 48

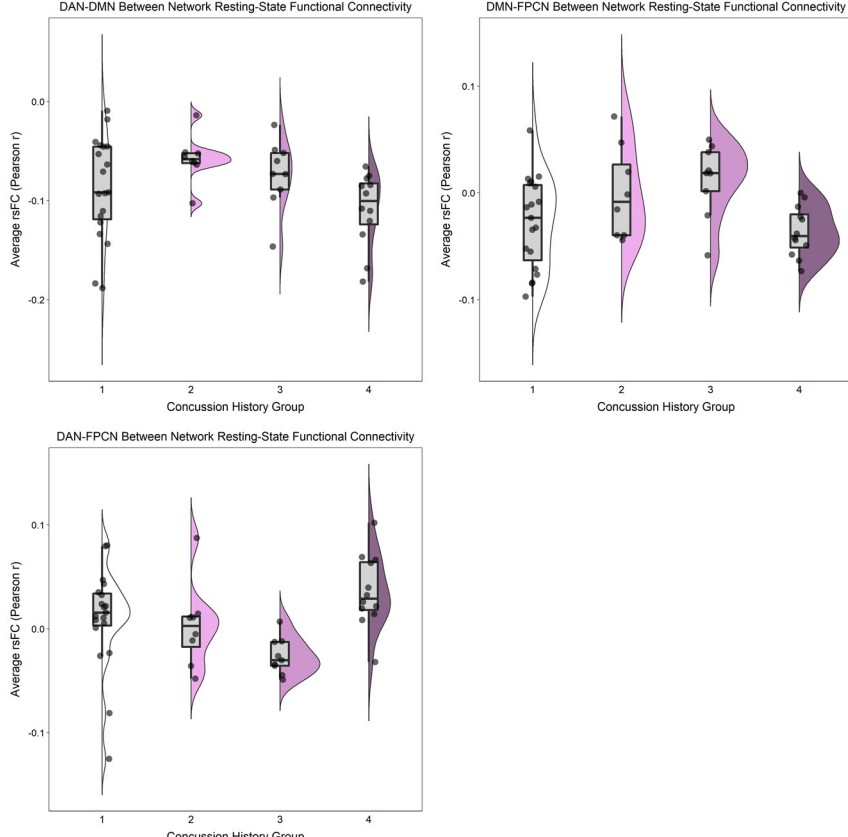

**Fig 4. Between-network resting-state functional connectivity (rsFC) in large-scale networks separated by four ordinal concussion history groups.** Between-network rsFC was calculated as the average Pearson *r* correlation (y-axes) between each of the individual ROIs from one specific network and all individual ROIs in another specific network across the time-series: Dorsal Attention Network and Default Mode Network (top-left); Default Mode Network and Frontoparietal Control Network (top-right); and Dorsal Attention Network and Frontoparietal Control Network (bottom). Individual points represent participant-level outcomes for each measure within each of the concussion history groups (1 = 0 or 1 concussion [*n* = 19]; 2 = 2 to 4 concussions [*n* = 8]; 3 = 5 to 7 concussions [*n* = 9]; and 4 = 8 or more concussions [*n* = 12]). Boxplots represent the median and interquartile range, and the whiskers extend to 1.5 times the interquartile range. The violin plot is a depiction of the density of individual rsFC values for each measure.

participants included in our analyses, and most of them identified as White/non-Hispanic. Lack of racial and ethnic diversity, as well as other potential determinants of brain health, limit the generalizability of our findings to the population. On the whole, our sample reports limited health conditions and functional limitations [42, 45]; however, there are thousands of former collegiate football players who may or may not be similar in their current health status. With this cross-sectional data, we cannot yet determine whether trajectories of brain health-related outcomes as players age are associated with accumulated concussion injuries or repetitive head impact exposures, and whether changes in brain health are different for former football players than non-football players. Future work should prospectively examine the time course of changes in functional connectivity in aging former football players and evaluate potential modifiers of these changes over time (i.e., expected brain health changes due to aging versus accelerated decline resulting from acquired brain trauma).

Among former collegiate football players at early midlife, we did not observe associations among concussion history or repetitive head impact exposure and neural signatures of altered

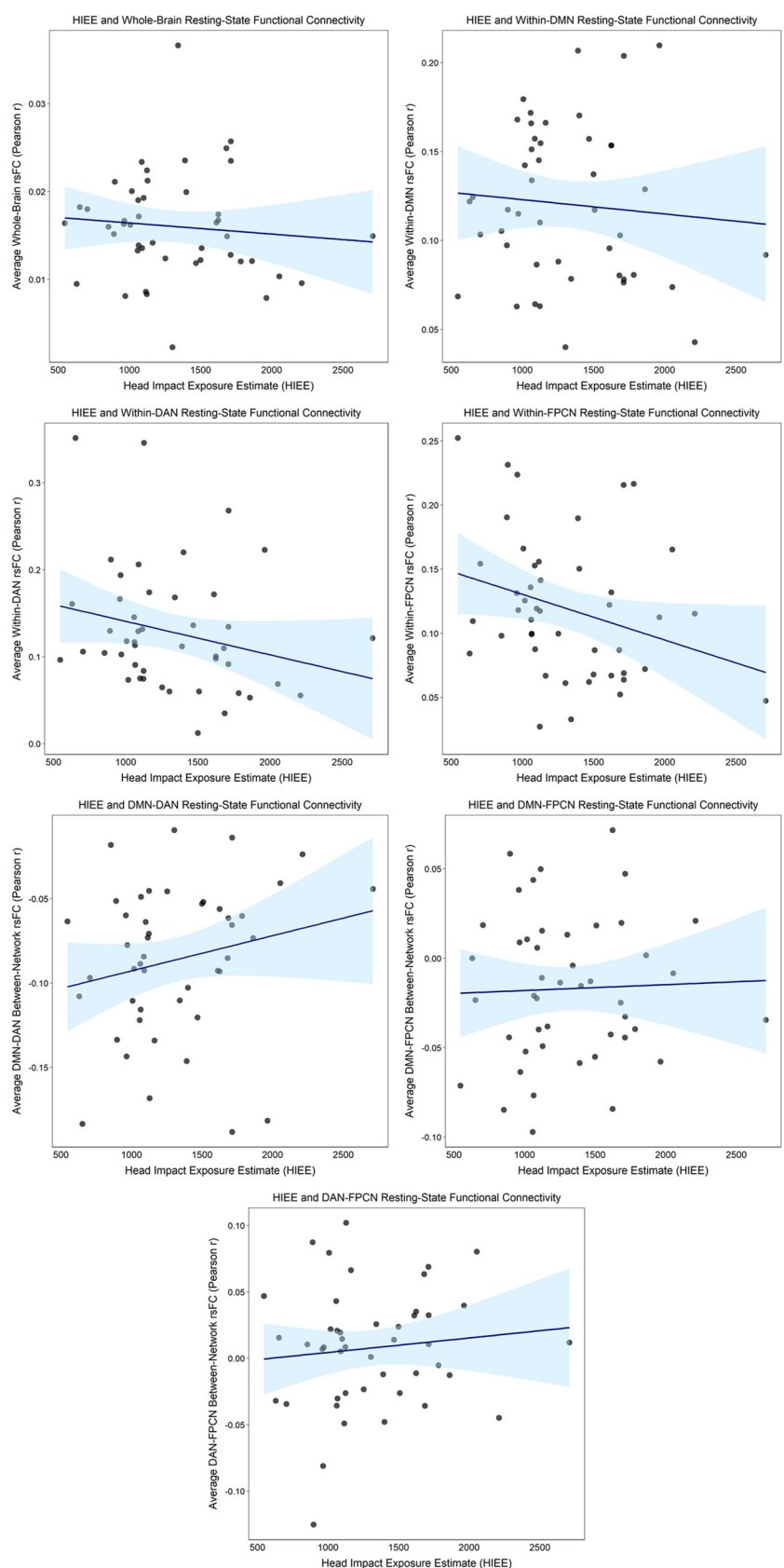

**Fig 5. Plots of head impact exposure estimates (HIEE) and average resting-state functional connectivity (rsFC) measures in large-scale networks.** Whole-brain rsFC was calculated as the average Pearson *r* correlation (y-axes) between each of the 300 regions of interest (ROIs) and all other ROIs across the time-series. Within-network rsFC for the Dorsal Attention Network (DAN), the Default Mode Network (DMN), and the Frontoparietal Control Network (FPCN) was calculated as the average Pearson *r* correlation between each of the individual ROIs from the given network and all other ROIs within that same network across the time-series. Between-network rsFC was calculated as the average Pearson *r* correlation between each of the individual ROIs from one specific network and all individual ROIs in another specific network across the time-series: DMN-DAN; DMN-FPCN; and DAN-FPCN. Individual points represent each of the individual participants.

large-scale functional network connectivity that have been observed in studies of age-related or clinically significant cognitive declines, as well as acute SRC. While SRC history and repetitive head impacts have been associated with changes in brain health and function in former football players and other collision and contact sport athletes, our preliminary data suggest that associations with functional network connectivity, if they exist, may not be detectable prior to older ages. Continuing to study these former players prospectively will be useful in identifying predispositions to potential accelerated aging processes, and the age at which they are most likely to present.

## Supporting information

**S1 File. Supplemental results.** Linear regressions for the effect of individual covariates on each of the resting-state functional connectivity (rsFC) outcomes described in body of the manuscript.
(DOCX)

## Acknowledgments

First, we would like to thank the athletes who participated in this study. We would also like to thank Candice Goerger, Robyn Furger, and Leah Thomas for their invaluable efforts in study coordination and management. Finally, we are also grateful for the many important contributions to this study made by Drs. Weili Lin, Hao Guan, and Andrew Nencka.

## Author Contributions

**Conceptualization:** Samuel R. Walton, Jacob R. Powell, Benjamin L. Brett, Michael A. McCrea, Kevin M. Guskiewicz, Kelly S. Giovanello.

**Data curation:** Samuel R. Walton, Jacob R. Powell, Weiyan Yin, Zachary Yukio Kerr, Kelly S. Giovanello.

**Formal analysis:** Samuel R. Walton, Jacob R. Powell, Weiyan Yin.

**Funding acquisition:** Zachary Yukio Kerr, Michael A. McCrea, Kevin M. Guskiewicz.

**Investigation:** Zachary Yukio Kerr, Michael A. McCrea, Kevin M. Guskiewicz, Kelly S. Giovanello.

**Methodology:** Samuel R. Walton, Benjamin L. Brett, Zachary Yukio Kerr, Mingxia Liu, Michael A. McCrea, Kevin M. Guskiewicz, Kelly S. Giovanello.

**Project administration:** Michael A. McCrea, Kevin M. Guskiewicz, Kelly S. Giovanello.

**Resources:** Michael A. McCrea, Kevin M. Guskiewicz, Kelly S. Giovanello.

**Software:** Weiyan Yin.

**Supervision:** Zachary Yukio Kerr, Michael A. McCrea, Kevin M. Guskiewicz, Kelly S. Giovanello.

**Visualization:** Samuel R. Walton.

**Writing – original draft:** Samuel R. Walton.

**Writing – review & editing:** Samuel R. Walton, Jacob R. Powell, Benjamin L. Brett, Weiyan Yin, Zachary Yukio Kerr, Mingxia Liu, Michael A. McCrea, Kevin M. Guskiewicz, Kelly S. Giovanello.

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
