## [Decision Letter · Decision Letter 0]

21 Mar 2022

PONE-D-22-05923Associations of Lifetime Concussion History and Repetitive Head Impact Exposure with Resting-State Functional Connectivity in Former Collegiate American Football Players: An NCAA 15-Year Follow-Up StudyPLOS ONE

Dear Dr. Walton,

Thank you for submitting your manuscript to PLOS ONE. After careful consideration, we feel that it has merit but does not fully meet PLOS ONE’s publication criteria as it currently stands. Therefore, we invite you to submit a revised version of the manuscript that addresses the points raised during the review process.

More specifically, Reviewer 1 brings up several points that I agree need to be addressed prior to consideration for publication. In terms of specific comments to pay attention to when revising your document is that thought of "reframing" the introduction and discussion of your manuscript as either "preliminary" or "baseline." Reviewers 1 and 2 each have provided meaningful feedback for your consideration as your authorship pursues the revision of this manuscript. As Reviewer 2 suggests, I believe this is an important manuscript, however, a more nuanced approach would be appreciated provided the sample size and potential attrition over time.   

We look forward to receiving your revised manuscript.

Kind regards,

Jacob Resch, Ph.D.

Academic Editor

PLOS ONE

Journal Requirements:

"We would like to thank the athletes who participated in this study as well as the

397 National Collegiate Athletic Association for its support of this work."

"KMG and MAM received funding for this study from the National Collegiate Athletic Association (https://www.ncaa.org/). The funders had no role in study design, data collection and analysis, decision to publish, or preparation of the manuscript."

Reviewers' comments:

Reviewer's Responses to Questions

**Comments to the Author**

1. Is the manuscript technically sound, and do the data support the conclusions?

Reviewer #1: No

Reviewer #2: Yes

2. Has the statistical analysis been performed appropriately and rigorously? 

Reviewer #1: Yes

Reviewer #2: Yes

3. Have the authors made all data underlying the findings in their manuscript fully available?

Reviewer #1: Yes

Reviewer #2: No

4. Is the manuscript presented in an intelligible fashion and written in standard English?

Reviewer #1: Yes

Reviewer #2: Yes

5. Review Comments to the Author

Reviewer #1: Walton and colleagues report data from a subset of former collegiate American football athletes who underwent in-person data collection through the 15-year follow up of the original NCAA study. This report focused on rs-fMRI data. Somewhat similar to other recent reports from this dataset, results suggested minimal support for an association between self-reported concussion history or repetitive head impact exposure and brain health, operationalized here as specific within- and between-network connectivity strengths.

The small N of this study makes it difficult to draw meaningful conclusions from the null findings. The authors generally did a good job of interpreting the data cautiously and pointing out the limitations of the study sample and design, but given these, I am on the fence about the appropriateness of this study as a standalone publication in its current form. Conceptually, I felt the introduction did not adequately set up the rationale for a study of presumably healthy men in their 30s and 40s. The hypothesis-driven approach is much appreciated, but I am not sure that data from older adult studies mostly focused on patients with MCI/dementia due to Alzheimer’s disease is ideal here. I was not convinced that we should have ever expected to find the hypothesized changes in this particular sample in the first place (young, healthy males) simply because they have varying degrees of prior head trauma exposure. This would have required a biomarker with a sensitivity to preclinical neurodegenerative pathophysiology that I don’t think exists anywhere. Even well-validated biomarkers of AD pathophysiology require relatively widespread AD pathology and, usually, symptomatic patients before they are clearly altered.

As the authors acknowledge, the real goal is longitudinal tracking. I am a little worried that the current study N is only 55 and further attrition is expected over time. I view this study as basically a report of baseline rs-fMRI data in a sample of former collegiate American football players that notes no clear association with prior head trauma, which by itself (i.e., an isolated biomarker without other biologic or clinical measurement) is not terribly compelling given the primary conclusion is essentially “we’ll now wait and see if things change over time.” The data may be better presented as something like a short report.

I thank the authors for beginning to organize the important data expected to come from the follow-up NCAA study and look forward to seeing future results from this unique cohort. I provide other comments and questions for the authors to consider:

1) L73-75: This opening statement is a little confusing. Why is MCI called out specifically (also no need to capitalize Mild Cognitive Impairment)? Also not clear what is meant by “typical aging.”

2) L80-82: Can the authors be more specific than “neurodegenerative changes”? I caution against using vague terminology (including “typical aging”). If you mean cognitive healthy older adults, or clinically normal older adults, etc., that would be preferred over “typical aging” in this context (applicable throughout the manuscript).

3) L85-86: Is rs-fMRI still considered a burgeoning method?

4) L97: MCI is not an example of a neurodegenerative disease. MCI is a cluster of symptoms potentially resulting from an underlying neurodegenerative disease. Alzheimer’s disease is one example of a neurodegenerative disease, which can manifest as symptoms that get classified as either MCI (objective cognitive/behavioral changes without impact on functional independence) or dementia (same as MCI, but now with loss of functional independence). Alzheimer’s disease is not synonymous with dementia, nor is it a more severe form of MCI. I strongly recommend modifying terminology throughout the paper to more accurately represent symptom-based/syndromic phenomenon (e.g., MCI, dementia) distinct from the neurodegenerative disease causing those symptoms (e.g., Alzheimer’s disease).

5) L107: I recommend saying “even without overt symptoms” instead of “injury.”

6) L108-109: Authors previously stated that rsFC changes in “those with MCI and AD are similar to changes observed with typical aging processes.” Concerns with terminology aside, it is unclear then what “alterations in functional connectivity that are similar to those observed in pathological aging” is referring to if rsFC changes are similar between healthy and unhealthy aging groups.

7) L244: While I would not personally consider multiple comparison adjustments a hard and fast rule, it would be helpful for the authors to provide rationale for an a priori alpha of p < .05 given the number of models, or include interpretation of alternate metrics to complement the p values.

8) L248-249: It is necessary to give readers a sense of the sample being studied here. How many athletes were contacted to participate in the in-person phase? How many outright declined compared to the 55 who were enrolled? Are there any metrics that can be provided to determine potential demographic/exposure differences between those who enrolled and those who declined? Were any clinical evaluations performed for these participants to gauge cognitive/behavioral status?

9) L277-280: Some of the adjusted R-squared values for these models are decently high. While not statistically significant, the standardized beta-weights for HIEE in a handful of the models are intriguing (e.g., within DAN and FPCN, between DMN-DAN). Given the low N for this study, and at least one of these associations being in the hypothesized direction (lower within DAN), it may be worth incorporating effect size estimates into your interpretation and also providing readers with a sense of study power in the methods (i.e., what effect size would have been required to be detected as statistically significant given your N?).

10) General points regarding the Discussion: It is exceedingly difficult to “prove the null” hypothesis and draw firm conclusions about associations between head trauma exposure and rsFC based on this study. First, I have concerns about the underlying conceptual model of aging/neurodegenerative disease considering this was a sample of men in their 30s and 40s and presumably all are cognitively healthy (there were no details provided about cognitive/behavioral health). Therefore, identifying rsFC changes depended on methodology being so exquisitely sensitive to pathological brain changes (if they existed) that it would detect them decades prior to symptom onset (should that ultimately occur). I don’t know that we can assume that. 2) The ordinal categorization of concussion history is better than the dichotomization, but there remain questions about self-report accuracy given that some studies show self-report numbers on the order of 10s (such as this study) and others show self-report numbers on the order of hundreds to thousands. 3) The limitations section is well thought out and transparent, though I worry that acknowledgment of the limitations alone is insufficient and wonder whether we can really draw meaningful conclusions in light of these limitations.

11) The choice of figure(s) is unclear. Why decide to show only the group comparisons (null) for the Low vs. High concussion hx groups rather than the ordinal characterization and/or scatterplots depicting the HIEE associations?

Reviewer #2: The authors present an important paper on brain health (based on functional connectivity imaging metrics) in former collegiate football players and this manuscript will make a valuable addition to the literature. One primary concern is the limited description of the participants and how this relates to the larger story. The recent TES NINDS statement (Katz 2021) suggests 5 years of collision sports is needed to reach some magical “threshold”, it could be really interesting to see if this population sample reaches that threshold especially given the results. Presumably, collegiate football players also participated in high school, so one would suspect that all participants herein meet the TES criteria. Similarly, did these participants continue sports participation post-college (either RHI or non-RHI sports) given the known benefits of exercise on broadly stated brain health. Further, there is a real concern about a healthy person selection bias in the Methods (e.g., can’t travel). Can the authors elaborate on how they addressed this limitation? Otherwise, it seems to be the opposite of the BU studies with their unhealthy person bias.

Minor comments

Abstract – suggest adding some simplistic demographics to the participants section.

Methods – The rationale for “one or more years” of college football needs to be provided. I would also suggest adding career duration (years played) in Table 2 (acknowledging HIEE is a better metric) to make it easier to compare to other studies.

I commend the authors for their transparency in Table 1 on the different scanners, did the authors perform any post-hoc comparisons to ensure the scanner switch at institution B didn’t influence the outcomes (while recognizing that “location” was an appropriate covariate).

Line #99 – “changes” – can this be directionally defined? From the literature, it seems that either higher or lower is “bad” which makes findings hard to compare and interpret.

Line #133 – Can you clarify “suspicion” of psychotic disorder? Was anyone removed based on this criterion?

Line #194 – a very old reference for the definition of a concussion. Why not use a more contemporary definition (e.g., 5th CIS, CARE, etc).

In the Discussion the authors comment on the lack of studies investigating the middle-aged former football player (line #333 +/-), however Iverson has a couple of studies in this area (CJSM 2021, Frontiers 2021, J Neurotrauma 2021) which address this issue and their inclusion would provide a more comprehensive discussion.

Similarly, there are some studies outside of football (e.g., rugby) which address midlife health in collision sports athletes (e.g., Hunzinger MSSE 2021; Van Patten 2021 Frontiers; Inversion 2021 Frontiers)

I commend the authors for their transparency in the Figures.

6. PLOS authors have the option to publish the peer review history of their article (what does this mean?). If published, this will include your full peer review and any attached files.

Reviewer #1: No

Reviewer #2: No

---

## [Author Response · Author response to Decision Letter 0]

20 May 2022

Reviewer #1: Walton and colleagues report data from a subset of former collegiate American football athletes who underwent in-person data collection through the 15-year follow up of the original NCAA study. This report focused on rs-fMRI data. Somewhat similar to other recent reports from this dataset, results suggested minimal support for an association between self-reported concussion history or repetitive head impact exposure and brain health, operationalized here as specific within- and between-network connectivity strengths.

The small N of this study makes it difficult to draw meaningful conclusions from the null findings. The authors generally did a good job of interpreting the data cautiously and pointing out the limitations of the study sample and design, but given these, I am on the fence about the appropriateness of this study as a standalone publication in its current form. Conceptually, I felt the introduction did not adequately set up the rationale for a study of presumably healthy men in their 30s and 40s. The hypothesis-driven approach is much appreciated, but I am not sure that data from older adult studies mostly focused on patients with MCI/dementia due to Alzheimer’s disease is ideal here. I was not convinced that we should have ever expected to find the hypothesized changes in this particular sample in the first place (young, healthy males) simply because they have varying degrees of prior head trauma exposure. This would have required a biomarker with a sensitivity to preclinical neurodegenerative pathophysiology that I don’t think exists anywhere. Even well-validated biomarkers of AD pathophysiology require relatively widespread AD pathology and, usually, symptomatic patients before they are clearly altered.

Author Response: We understand the reviewer’s concerns and have made changes throughout the manuscript to better present this study as a preliminary investigation of a potential objective marker of declining brain health. Specifically, language has been revised, including in the hypotheses, to frame our study purpose through an inductive lens—that is, if long-term changes in functional connectivity exists in those with history of one or more mild TBI and/or exposure to repetitive head trauma, we hypothesize that rsFC in this sample would appear to be similar to connectivity patterns observed in older individuals. This was our original study purpose, and the reviewer’s feedback has been very helpful in recognizing where it was not well communicated.

As the authors acknowledge, the real goal is longitudinal tracking. I am a little worried that the current study N is only 55 and further attrition is expected over time. I view this study as basically a report of baseline rs-fMRI data in a sample of former collegiate American football players that notes no clear association with prior head trauma, which by itself (i.e., an isolated biomarker without other biologic or clinical measurement) is not terribly compelling given the primary conclusion is essentially “we’ll now wait and see if things change over time.” The data may be better presented as something like a short report.

Author Response: In line with our response to the preceding comment, we have revised the language in this paper to better present this as a preliminary report of rsFC in this sample. We have also acknowledged the recent work with a heavily overlapping sample detailing the general health and well-being of the sample via objective and subjective measures of cognitive and neurobehavioral functioning. 

I thank the authors for beginning to organize the important data expected to come from the follow-up NCAA study and look forward to seeing future results from this unique cohort. I provide other comments and questions for the authors to consider:

1) L73-75: This opening statement is a little confusing. Why is MCI called out specifically (also no need to capitalize Mild Cognitive Impairment)? Also not clear what is meant by “typical aging.”

Author Response: This comment was very helpful, thank you. MCI was called out specifically as it has been a focus of recent work in former football players, and it has been associated with concussion history and repetitive head impacts in those players. We have changed our opening statement to use MCI as an example of abnormal cognitive decline, and the capitalization has been removed. This change was also in line with other comments from this reviewer to use more specific terminology rather than “typical aging”. We have made similar changes throughout the manuscript. 

2) L80-82: Can the authors be more specific than “neurodegenerative changes”? I caution against using vague terminology (including “typical aging”). If you mean cognitive healthy older adults, or clinically normal older adults, etc., that would be preferred over “typical aging” in this context (applicable throughout the manuscript).

Author Response: Similar to the reviewer’s previous comment, this was helpful feedback. Changes have been made in the introduction and throughout the manuscript to be less vague regarding these important points. 

3) L85-86: Is rs-fMRI still considered a burgeoning method?

Author Response: The word burgeoning has been removed, and the statement now reads: “One method for examining biologic changes in brain health is resting-state functional activity as measured with magnetic resonance imaging (MRI).”

4) L97: MCI is not an example of a neurodegenerative disease. MCI is a cluster of symptoms potentially resulting from an underlying neurodegenerative disease. Alzheimer’s disease is one example of a neurodegenerative disease, which can manifest as symptoms that get classified as either MCI (objective cognitive/behavioral changes without impact on functional independence) or dementia (same as MCI, but now with loss of functional independence). Alzheimer’s disease is not synonymous with dementia, nor is it a more severe form of MCI. I strongly recommend modifying terminology throughout the paper to more accurately represent symptom-based/syndromic phenomenon (e.g., MCI, dementia) distinct from the neurodegenerative disease causing those symptoms (e.g., Alzheimer’s disease).

Author Response: Thank you for this clarification. We have made efforts to better distinguish these important characteristics in the language used throughout the paper, in line with the reviewer’s previous comments as well. 

5) L107: I recommend saying “even without overt symptoms” instead of “injury.”

Author Response: Great suggestion. We have changed this statement accordingly: “Further evidence suggests that rsFC changes also occur in relation to repetitive head impacts, even without overt clinical signs or symptoms (e.g., concussion).” 

6) L108-109: Authors previously stated that rsFC changes in “those with MCI and AD are similar to changes observed with typical aging processes.” Concerns with terminology aside, it is unclear then what “alterations in functional connectivity that are similar to those observed in pathological aging” is referring to if rsFC changes are similar between healthy and unhealthy aging groups.

Author Response: This is helpful feedback. We have modified the final section of the introduction to address this comment as well as to help address the concern noted in other reviewer comments about the underlying conceptual framework for this study. 

The reviewer’s comment that, “[a]uthors previously stated that rsFC changes in ‘those with MCI and AD are similar to changes observed with typical aging processes,’” is important, and we have attempted to qualify that initial statement by indicating that changes in the presence of MCI and AD diagnoses suggest an exacerbated or accelerated process of functional changes when compared to normal age-related changes, theoretically relating to accelerated biological aging. 

We have changed the statement, “alterations in functional connectivity that are similar to those observed in pathological aging,” to address the notion that the functional connectivity changes exist in normal aging but may appear in younger individuals: “Taken together, long-term alterations in functional connectivity may result from greater exposure to SRC and repetitive head impacts, even in relatively young individuals, and these changes may resemble those observed in individuals experiencing cognitive decline. In this light, it is also reasonable to consider functional connectivity changes as potential biomarkers for the advanced aging and the early onset of pathological changes in former football players.” 

7) L244: While I would not personally consider multiple comparison adjustments a hard and fast rule, it would be helpful for the authors to provide rationale for an a priori alpha of p < .05 given the number of models, or include interpretation of alternate metrics to complement the p values.

Author Response: This is a fair point. We have added language about the preliminary nature of this study and to acknowledge that we will interpret the results based on both p-values and measures of effect size (standardized beta values). We have also included estimates of post hoc power for each regression model in the supplement (S1). 

8) L248-249: It is necessary to give readers a sense of the sample being studied here. How many athletes were contacted to participate in the in-person phase? How many outright declined compared to the 55 who were enrolled? Are there any metrics that can be provided to determine potential demographic/exposure differences between those who enrolled and those who declined? Were any clinical evaluations performed for these participants to gauge cognitive/behavioral status?

Author Response: Thank you, this is a great suggestion. We have included the available information regarding recruitment and retention of study participants in the methods and results sections. Additionally, both objective and subjective measures of neurobehavioral function were reported in a recently published paper, so we’ve cited that paper in the results section accompanying a statement that the generally fell within the average range across indices of neurobehavioral function. 

9) L277-280: Some of the adjusted R-squared values for these models are decently high. While not statistically significant, the standardized beta-weights for HIEE in a handful of the models are intriguing (e.g., within DAN and FPCN, between DMN-DAN). Given the low N for this study, and at least one of these associations being in the hypothesized direction (lower within DAN), it may be worth incorporating effect size estimates into your interpretation and also providing readers with a sense of study power in the methods (i.e., what effect size would have been required to be detected as statistically significant given your N?).

Author Response: This is an important suggestion. We have included interpretations of the standardized beta-weights in the paper as well as post hoc power estimates in the paper and supplemental file. 

10) General points regarding the Discussion: It is exceedingly difficult to “prove the null” hypothesis and draw firm conclusions about associations between head trauma exposure and rsFC based on this study. First, I have concerns about the underlying conceptual model of aging/neurodegenerative disease considering this was a sample of men in their 30s and 40s and presumably all are cognitively healthy (there were no details provided about cognitive/behavioral health). Therefore, identifying rsFC changes depended on methodology being so exquisitely sensitive to pathological brain changes (if they existed) that it would detect them decades prior to symptom onset (should that ultimately occur). I don’t know that we can assume that. 2) The ordinal categorization of concussion history is better than the dichotomization, but there remain questions about self-report accuracy given that some studies show self-report numbers on the order of 10s (such as this study) and others show self-report numbers on the order of hundreds to thousands. 3) The limitations section is well thought out and transparent, though I worry that acknowledgment of the limitations alone is insufficient and wonder whether we can really draw meaningful conclusions in light of these limitations.

Author Response: Thank you for this discussion. It relates to prior comments from this reviewer as well, and we have attempted to address key components of this in the revised manuscript. Specifically: 1— We have provided more details regarding the conceptual model. We have additionally provided a statement about, and citation for, the participants’ current cognitive/neurobehavioral health. Further, we have more clearly stated that this is a preliminary study throughout the manuscript; 2¬—Self-reported concussion history is indeed a notable consideration, and we’ve added text to address this in the body of the discussion rather than just in the limitations; 3—We have hopefully addressed the reviewer’s concerns in response to the first two points. 

11) The choice of figure(s) is unclear. Why decide to show only the group comparisons (null) for the Low vs. High concussion hx groups rather than the ordinal characterization and/or scatterplots depicting the HIEE associations?

Author Response: The choice to present one set of graphs was to give a visual example of the spread of the data in relation to a predictor of interest. Since all the associations we tested were not statistically significant, we opted for what we felt to be the simplest depiction (Low vs. High concussion history). However, after considering the reviewer’s comment, we have included graphical representations of the associations between both concussion history categorizations and rsFC values as well as between HIEE and rsFC values. There are now 5 total figures included in the paper. 

Reviewer #2: The authors present an important paper on brain health (based on functional connectivity imaging metrics) in former collegiate football players and this manuscript will make a valuable addition to the literature. One primary concern is the limited description of the participants and how this relates to the larger story. The recent TES NINDS statement (Katz 2021) suggests 5 years of collision sports is needed to reach some magical “threshold”, it could be really interesting to see if this population sample reaches that threshold especially given the results. Presumably, collegiate football players also participated in high school, so one would suspect that all participants herein meet the TES criteria. Similarly, did these participants continue sports participation post-college (either RHI or non-RHI sports) given the known benefits of exercise on broadly stated brain health. Further, there is a real concern about a healthy person selection bias in the Methods (e.g., can’t travel). Can the authors elaborate on how they addressed this limitation? Otherwise, it seems to be the opposite of the BU studies with their unhealthy person bias.

Author Response: Thank you for bringing up this important context. We have added both total years of football play as well as the proportion who played professional football to Table 2 alongside other demographic data. Beyond participation in professional football, we do not possess data on post-collegiate participation in other sports or other types of physical activity. 

Regarding the “healthy person bias”, it would be hard to say that this was definitely the case with our sample. In the table below, we have data from a separate manuscript which is currently in review elsewhere (please keep these data confidential) on rates of lower-than-average function/performance. The data indicate that not all our participants were within normal limits. Our participants may also have been interested in the current state of their personal health and/or concerned with their long-term brain health, which could have been the driving factor for their choice to participate in this longitudinal follow-up. We did not directly measure this, and therefore cannot account for personal reasons for engaging in our study. Relatedly, in response to a comment by another reviewer, we did add text and an accompanying citation to the results section (Participants) stating that, “…participants were relatively healthy with regard to performance-based and self-reported measures of cognitive function as described previously by Brett et al.[44]” Finally, we tested the hypothesis that exposure leads to differences in an outcome (rsFC). There was a significant range (extending to the upward limit) in each of our exposure variables, suggesting that if more concussions and RHI would associate with differences in rsFC, we would in theory observe that association here.

Table 4.

Proportion of scores equal to or exceeding one standard deviation on objective and subjective measures. 

 Neuro-QoL Cognition BRIEF-A MI BRIEF-A BRI

 Sample (N = 57) WNL Low WNL Low WNL Low

Objective Cognitive Functioning n (%) n (%) n (%) n (%) n (%) n (%) n (%)

 HVLT-R Immediate Recall a WNL 45 (78.9) 42 (93.3) 3 (6.7) — — — —

 Low 12 (21.0) 10 (83.3) 2 (16.7) — — — —

 HVLT- R Delayed Recall a WNL 43 (75.4) 42 (97.7) 1 (2.3) — — — —

 Low 14 (24.6) 10 (71.4) 4 (28.6) — — — —

 SDMT a WNL 55 (96.5) 50 (90.9) 5 (9.1) — — — —

 Low 2 (3.5) 2 (100) 0 (0) — — — —

 TMT-A a WNL 38 (66.7) 34 (89.5) 4 (10.5) — — — —

 Low 19 (33.3) 18 (94.7) 1 (5.3) — — — —

 TMT-B a WNL 34 (59.6) 30 (88.2) 4 (11.8) 27 (79.4) 7 (20.6) 27 (79.4) 7 (20.6)

 Low 23 (40.4) 22 (95.7) 5 (4.3) 21 (91.3) 2 (8.7) 19 (82.6) 4 (17.4)

 F-A-S a WNL 50 (87.7) 48 (96.0) 2 (4.0) 44 (88.0) 6 (12.0) 42 (84.0) 8 (16.0)

 Low 7 (12.3) 4 (57.1) 3 (42.9) 4 (57.1) 3 (42.9) 4 (57.1) 3 (42.9)

Subjective Cognitive Functioning 

 Neuro-QoL Cognition a WNL 52 (91.2) — — — — — —

 Low 5 (8.8) — — — — — —

 BRIEF-MI b WNL 48 (84.2) 46 (95.8) 2 (4.2) — — — —

 Low 9 (15.8) 6 (66.7) 3 (33.3) — — — —

 BRIEF-BRI b WNL 46 (80.7) 44 (95.7) 2 (4.3) — — — —

 Low 11 (19.3) 8 (72.7) 3 (27.3) — — — —

Note. Low = lower than average function/performance (≥ 1 SD); BRI = Behavioral Regulation Index; BRIEF-A = Behavior Rating Inventory of Executive Function – Adult; BSI-18 GSI= Brief Symptom Inventory-18 Global Severity Index; F-A-S = Verbal Fluency; HVLT-R = Hopkins Verbal Learning Test-Revised; MI = Metacognition Index; Neuro-QoL Cognition = Quality of Life in Neurological Disorders Cognitive Functioning Short-form; SDMT = Symbol Digit Modalities Test; TMT = Trail Making Test; WNL = within normal limits (± 1 SD).

a = Low defined as ≤1 SD of the mean.

b = Low defined as ≥1 SD of mean.

Minor comments

Abstract – suggest adding some simplistic demographics to the participants section.

Author Response: Good suggestion. In addition to age and gender, we have also added the proportion identifying as white/Caucasian and the average total number of years playing football. Concussion histories are detailed later in the abstract. 

Methods – The rationale for “one or more years” of college football needs to be provided. I would also suggest adding career duration (years played) in Table 2 (acknowledging HIEE is a better metric) to make it easier to compare to other studies.

Author Response: Thank you for these suggestions. We have added years of football play to Table 2 as well as the proportion of participants who went on to play professional football after college. We have edited the Participants paragraph of the Materials & Methods section to detail inclusion and exclusion criteria for our study. The rationale for one or more years of college football play was that this was a study of former collegiate football players, and they needed to have this experience in order to be included in the study. 

I commend the authors for their transparency in Table 1 on the different scanners, did the authors perform any post-hoc comparisons to ensure the scanner switch at institution B didn’t influence the outcomes (while recognizing that “location” was an appropriate covariate).

Author Response: The reviewer brings up a reasonable suggestion. There were no statistically significant differences between the two protocols at Institution B for any of the seven study outcomes (ps ≥ .059). 

Line #99 – “changes” – can this be directionally defined? From the literature, it seems that either higher or lower is “bad” which makes findings hard to compare and interpret.

Author Response: This is a great point. In response to this comment as well as those from the other reviewer, we have tried to clarify this instance and other similar vague statements throughout the introduction and discussion. 

Line #133 – Can you clarify “suspicion” of psychotic disorder? Was anyone removed based on this criterion?

Author Response: We have removed the term “suspicion” from this statement and have included a statement prior that all potential participants were screened for the presence of psychotic disorder with active symptoms as part of the recruitment process. We do not have a specific number to report regarding how many people were screened out as a result of that particular exclusion criterion. 

Line #194 – a very old reference for the definition of a concussion. Why not use a more contemporary definition (e.g., 5th CIS, CARE, etc).

Author Response: This is a good point from the reviewer. Provided that this study was a longitudinal follow-up from a study performed 15 years prior, we wanted to keep the same definition of concussion across time points to ensure that participants were being asked to report their lifetime concussion history using a consistent measure. We acknowledge that self-reported concussion history, in general, has flaws in both the early part of the discussion section and the limitations paragraph. 

In the Discussion the authors comment on the lack of studies investigating the middle-aged former football player (line #333 +/-), however Iverson has a couple of studies in this area (CJSM 2021, Frontiers 2021, J Neurotrauma 2021) which address this issue and their inclusion would provide a more comprehensive discussion.

Similarly, there are some studies outside of football (e.g., rugby) which address midlife health in collision sports athletes (e.g., Hunzinger MSSE 2021; Van Patten 2021 Frontiers; Inversion 2021 Frontiers)

Author Response: This is a good point, and these are valuable studies to the understanding of brain health in former athletes. Thank you for providing this perspective. We have clarified our statement in the paper to say, “Specifically, objective markers of biological brain health in former athletes below 50 years of age are relatively understudied compared to their older counterparts.” This is the point that was originally intended, and we believe the revised statement better reflects that intent. 

I commend the authors for their transparency in the Figures.

Author Response: Thank you very much. In response to a comment from the other reviewer regarding the selection of a single set of figures (dichotomous concussion history), we have also included figures depicting the outcome rsFC data in relation to the 4-category concussion history as well as HIEE.

---

## [Decision Letter · Decision Letter 1]

16 Jun 2022

PONE-D-22-05923R1Associations of Lifetime Concussion History and Repetitive Head Impact Exposure with Resting-State Functional Connectivity in Former Collegiate American Football Players: An NCAA 15-Year Follow-Up StudyPLOS ONE

Dear Dr. Walton,

Thank you for submitting your manuscript to PLOS ONE. After careful consideration, we feel that it has merit but does not fully meet PLOS ONE’s publication criteria as it currently stands. Therefore, we invite you to submit a revised version of the manuscript that addresses the points raised during the review process.

More specifically, I would ask that you and the authorship pay close attention to Reviewer 1's comments regarding the clarification of definitions used for neurodegenerative disease. Reviewer 1's thoughtful comments will assist the readership in further understanding what your findings are and what they are not. Reviewer 2's comments will also help craft a better manuscript in terms of readability and a more thoughtful discussion. 

We look forward to receiving your revised manuscript.

Kind regards,

Jacob Resch, Ph.D.

Academic Editor

PLOS ONE

Journal Requirements:

Reviewers' comments:

Reviewer's Responses to Questions

**Comments to the Author**

1. If the authors have adequately addressed your comments raised in a previous round of review and you feel that this manuscript is now acceptable for publication, you may indicate that here to bypass the “Comments to the Author” section, enter your conflict of interest statement in the “Confidential to Editor” section, and submit your "Accept" recommendation.

Reviewer #1: (No Response)

Reviewer #2: (No Response)

2. Is the manuscript technically sound, and do the data support the conclusions?

Reviewer #1: Yes

Reviewer #2: Yes

3. Has the statistical analysis been performed appropriately and rigorously? 

Reviewer #1: Yes

Reviewer #2: Yes

4. Have the authors made all data underlying the findings in their manuscript fully available?

Reviewer #1: No

Reviewer #2: Yes

5. Is the manuscript presented in an intelligible fashion and written in standard English?

Reviewer #1: Yes

Reviewer #2: Yes

6. Review Comments to the Author

Reviewer #1: Walton et al have significantly improved the organization and conceptual framework for this preliminary study. My remaining suggestions largely surround continued need for clarifying use of the terms MCI, dementia, AD, etc. in the paper (I commend the attempted accommodations already and am sympathetic to terminology being less-than-intuitive outside of aging/dementia specializations). Otherwise, I have no further content-specific recommendations.

As an additional point, I support the authors decision to appropriately steer clear of attempting to incorporate a TES framework or discussion into the current paper. The other reviewer indicated that “one would suspect that all participants herein meet the TES criteria,” which is not the case given that this study sample does not have evidence of a neurodegenerative disease and would not fulfill almost any of the TES criteria beyond having played 5 or more years of American football. As this sample continues to age, this is of course a worthy consideration and the data collected from these preliminary studies while still in their 30s and 40s could prove invaluable.

1) There remains confusion with the use of MCI, dementia, Alzheimer’s disease, etc. As stated in the prior review, MCI and dementia are clinically-defined entities with many different causes. One potential cause of MCI and dementia is Alzheimer’s disease. It would be more appropriate to simply open the introduction with “Several factors increase the risk for clinically significant cognitive decline in aging individuals (e.g., mild cognitive impairment [MCI] or dementia). If your specific hypotheses are around Alzheimer’s disease as an etiology for the MCI/dementia, or its associations with head trauma, then perfectly fine to call out AD specifically. Otherwise, probably better to just broadly mention the broad clinical syndrome categories (MCI, dementia) that reflect the presence of an underlying neurodegenerative disease (whether AD or something else).

2) L101-102: The terminology is off here. I suggest saying “…among individuals with age-related cognitive changes as well as clinically significant decline (e.g., MCI or dementia).” AD is not a more severe version of MCI and is not synonymous with dementia. It is one potential CAUSE of MCI/dementia. The cause of many adults’ MCI is Alzheimer’s disease (and, many cognitively normal older adults have Alzheimer’s disease but are resilient to the underlying brain changes, for many reasons). Please check the rest of the manuscript for instances where phrases like “MCI and AD” are used and modify accordingly.

3) L110: Not sure what “expected age-related functional declines…” refers to. Does this mean “functional” in the sense of rsFC changes? Or actual changes in daily function? The latter is not a part of normal or healthy aging.

4) L113: Consider “…individuals with clinically significant cognitive decline may be experiencing exacerbated or accelerated…” (similar suggestion throughout)

5) Author Response: Great suggestion. We have changed this statement accordingly: “Further evidence suggests that rsFC changes also occur in relation to repetitive head impacts, even without overt clinical signs or symptoms (e.g., concussion).”

Reviewer response: a little confusing as worded since at first glance it seems like you are saying that this sentence is describing an example of (“e.g.”) what a concussion is. Consider “…even without the overt clinical signs or symptoms consistent with a concussion diagnosis.”

Reviewer #2: I commend the authors for their response and revision of their manuscript to reflect the concerns of the two reviewers and I am now generally supportive of publication. However, a few smaller comments/considerations remain.

In response to reviewer #1, it is worth noting that cognitive and behavioral deficits (including CTE – e.g., Chris Henry died at 26 with CTE and cognitive/behavioral deficits) have been identified in this age group and this is probably worth noting in the introduction.

It might be simply a formatting issue on the track changes version, but the “burgeoning” sentence is currently a one sentence paragraph.

In regard to the Table in the response, unfortunately the formatting from the journal to the reviewer makes this largely unreadable; however, the text response explains. However, this raises an interesting point regarding the below average characteristics. One certainly understands the unfortunate need for salami science in the current environment, but in this case it really weakens this paper. The ability to link and compare the rsfMRI data to the cognitive/behavioral data could have been a real strength especially if the below average functional “group” had poorer fMRI outcomes.

The fact that the average participant had ~12.5 years of playing experience is a real strength of the study and suggests these are players who likely started in grade school and played to/through college. This is the critical question, in my opinion, from a public health perspective. The NFL reflects so few people as compared to youth through high school and the lack of findings here (in view of reviewer’s #1 comments on extrapolation) is noteworthy and, in my opinion, warrants stronger commentary in the Discussion. Certainly this does not fully answer the public health question, but it contributes to the discussion in a meaningful way.

The only “concern” that remains is the ability to identify how many participants failed their screening – if the data isn’t available for this specific reason, then can the authors provide an overall number of participants who failed their screening for any reason? Surely this would have tracked for IRB purposes. This would alleviate (or perhaps exacerbate) the healthy person bias concern.

7. PLOS authors have the option to publish the peer review history of their article (what does this mean?). If published, this will include your full peer review and any attached files.

Reviewer #1: No

Reviewer #2: No

---

## [Author Response · Author response to Decision Letter 1]

31 Jul 2022

Dear Dr. Resch, 

 We want to thank you and the reviewers for your continued review of our work, and for providing feedback that will help strengthen the paper. Below we have provided detailed responses to each of the reviewer comments. Of note, the reviewers appear to disagree on the importance of discussing criteria for Traumatic Encephalopathy Syndrome, and similarly “cognitive and behavioral deficits” in this cohort (and general age group). We do not feel it is appropriate to discuss this in the present paper as it was not within the scope of the research question—relating to imaging biomarkers of brain health—and that discussion of such may lead readers to believe that we are insinuating that these are measurable through fMRI studies. In the context of comments provided by “Reviewer 1” previously, clear rsFC biomarkers for well-known pathologies like Alzheimer’s Disease are pre-clinical (or non-existent) and we would prefer to avoid any suggestion that we were attempting to observe one for TES. These are indeed important research questions, with powerful clinical implications, but are outside the scope of the present study and not feasibly addressed with our present data. We are open to further comments and discussion from the Editor if deemed necessary. 

Thank you again, 

Samuel R. Walton, PhD

Reviewer #1: Walton et al have significantly improved the organization and conceptual framework for this preliminary study. My remaining suggestions largely surround continued need for clarifying use of the terms MCI, dementia, AD, etc. in the paper (I commend the attempted accommodations already and am sympathetic to terminology being less-than-intuitive outside of aging/dementia specializations). Otherwise, I have no further content-specific recommendations.

As an additional point, I support the authors decision to appropriately steer clear of attempting to incorporate a TES framework or discussion into the current paper. The other reviewer indicated that “one would suspect that all participants herein meet the TES criteria,” which is not the case given that this study sample does not have evidence of a neurodegenerative disease and would not fulfill almost any of the TES criteria beyond having played 5 or more years of American football. As this sample continues to age, this is of course a worthy consideration and the data collected from these preliminary studies while still in their 30s and 40s could prove invaluable.

AUTHOR RESPONSE: We thank the reviewer for their continued review and support of our decision to not discuss TES criteria specifically in this sample. 

1) There remains confusion with the use of MCI, dementia, Alzheimer’s disease, etc. As stated in the prior review, MCI and dementia are clinically-defined entities with many different causes. One potential cause of MCI and dementia is Alzheimer’s disease. It would be more appropriate to simply open the introduction with “Several factors increase the risk for clinically significant cognitive decline in aging individuals (e.g., mild cognitive impairment [MCI] or dementia). If your specific hypotheses are around Alzheimer’s disease as an etiology for the MCI/dementia, or its associations with head trauma, then perfectly fine to call out AD specifically. Otherwise, probably better to just broadly mention the broad clinical syndrome categories (MCI, dementia) that reflect the presence of an underlying neurodegenerative disease (whether AD or something else).

AUTHOR RESPONSE: These suggestions are well received and have been incorporated as recommended. We do not have AD-specific hypotheses and any mention of AD specifically was meant to acknowledge the content of the cited works. We have made efforts throughout to remove specific mentions of AD where it could potentially be misleading to the reader. 

2) L101-102: The terminology is off here. I suggest saying “…among individuals with age-related cognitive changes as well as clinically significant decline (e.g., MCI or dementia).” AD is not a more severe version of MCI and is not synonymous with dementia. It is one potential CAUSE of MCI/dementia. The cause of many adults’ MCI is Alzheimer’s disease (and, many cognitively normal older adults have Alzheimer’s disease but are resilient to the underlying brain changes, for many reasons). Please check the rest of the manuscript for instances where phrases like “MCI and AD” are used and modify accordingly.

AUTHOR RESPONSE: This comment as well as the following two, which are similar, has helped us address these multiple instances of unclear nomenclature. We have made changes throughout to avoid imprecise words like “expected”, “atypical”, etc. and have opted to use language as suggested by the reviewer in this comment. 

3) L110: Not sure what “expected age-related functional declines…” refers to. Does this mean “functional” in the sense of rsFC changes? Or actual changes in daily function? The latter is not a part of normal or healthy aging.

AUTHOR RESPONSE: See the response to item ‘2)’ above.

4) L113: Consider “…individuals with clinically significant cognitive decline may be experiencing exacerbated or accelerated…” (similar suggestion throughout)

AUTHOR RESPONSE: See the response to item ‘2)’ above.

5) Author Response: Great suggestion. We have changed this statement accordingly: “Further evidence suggests that rsFC changes also occur in relation to repetitive head impacts, even without overt clinical signs or symptoms (e.g., concussion).”

Reviewer response: a little confusing as worded since at first glance it seems like you are saying that this sentence is describing an example of (“e.g.”) what a concussion is. Consider “…even without the overt clinical signs or symptoms consistent with a concussion diagnosis.”

AUTHOR RESPONSE: Thank you for helping to clarify this statement. The reviewer’s suggestion has been incorporated. 

Reviewer #2: I commend the authors for their response and revision of their manuscript to reflect the concerns of the two reviewers and I am now generally supportive of publication. However, a few smaller comments/considerations remain.

AUTHOR RESPONSE: We thank the reviewer for their continued efforts in reviewing this manuscript and for providing meaningful feedback. 

In response to reviewer #1, it is worth noting that cognitive and behavioral deficits (including CTE – e.g., Chris Henry died at 26 with CTE and cognitive/behavioral deficits) have been identified in this age group and this is probably worth noting in the introduction.

It might be simply a formatting issue on the track changes version, but the “burgeoning” sentence is currently a one sentence paragraph.

AUTHOR RESPONSE: We agree with the reviewer that cognitive and behavioral deficits in relatively young former collision sport athletes is concerning, especially as there may be associations with neuropathological changes in the brain. As we discussed above, we feel that it would be inappropriate to specifically call out to CTE and/or TES in the present paper as recognition of these issues is beyond the scope of this paper. Similarly, and in response to Reviewer #1 above, we have also removed mention of other potentially misleading statements including specific conditions (such as Alzheimer’s Disease) to avoid confusion for the reader. We did not aim to study clinical diagnoses in the present paper, and have left presentation of these conditions to more general terms—for example, “Studies have suggested that former football players may have earlier onset of, and/or increased risk for, cognitive decline or neurodegenerative disease diagnoses compared with the general population, purportedly due to their exposure to repetitive head impacts—SRC or otherwise—during football play.[4–6,8].” 

In regard to the Table in the response, unfortunately the formatting from the journal to the reviewer makes this largely unreadable; however, the text response explains. However, this raises an interesting point regarding the below average characteristics. One certainly understands the unfortunate need for salami science in the current environment, but in this case it really weakens this paper. The ability to link and compare the rsfMRI data to the cognitive/behavioral data could have been a real strength especially if the below average functional “group” had poorer fMRI outcomes.

AUTHOR RESPONSE: The concern with having no clinical data (e.g., cognitive test performance) is an important consideration. In the present study, we based our hypotheses and analyses on previous work that focused on how resting-state functional connectivity patterns are associated with age-related changes and changes observed in following head trauma—which may or may not co-occur with cognitive changes. While resting-state networks have been correlated with specific domains of cognitive testing outcomes in prior studies, these relationships are not well established and we did not feel that it would be appropriate to layer these additional exploratory analyses on top of this preliminary investigation. 

The fact that the average participant had ~12.5 years of playing experience is a real strength of the study and suggests these are players who likely started in grade school and played to/through college. This is the critical question, in my opinion, from a public health perspective. The NFL reflects so few people as compared to youth through high school and the lack of findings here (in view of reviewer’s #1 comments on extrapolation) is noteworthy and, in my opinion, warrants stronger commentary in the Discussion. Certainly this does not fully answer the public health question, but it contributes to the discussion in a meaningful way.

AUTHOR RESPONSE: Thank you for this suggestion. We have added the following statements in the fourth paragraph of the discussion section: “It is notable that only a few (n = 7) of the participants in this study played football after their collegiate careers while also reporting 12.5 years of football play, on average. This sample is therefore mostly representative of amateur athletes who began playing football at youth levels.”

The only “concern” that remains is the ability to identify how many participants failed their screening – if the data isn’t available for this specific reason, then can the authors provide an overall number of participants who failed their screening for any reason? Surely this would have tracked for IRB purposes. This would alleviate (or perhaps exacerbate) the healthy person bias concern.

AUTHOR RESPONSE: Unfortunately, the tracking data for this study did not allow us to parse out who met which exclusion criteria vs. those who opted not to participate (via communication with the study team or via non-response to inquiry). Reasons for choosing not to participate and choosing not to respond were not reported, but could have been due to health-related concerns about travel and/or study participation. At this time, we cannot provide more information than has presently been given in the manuscript.

---

## [Decision Letter · Decision Letter 2]

18 Aug 2022

Associations of Lifetime Concussion History and Repetitive Head Impact Exposure with Resting-State Functional Connectivity in Former Collegiate American Football Players: An NCAA 15-Year Follow-Up Study

PONE-D-22-05923R2

Dear Dr. Walton,

We’re pleased to inform you that your manuscript has been judged scientifically suitable for publication and will be formally accepted for publication once it meets all outstanding technical requirements.

Kind regards,

Jacob Resch, Ph.D.

Academic Editor

PLOS ONE

Additional Editor Comments (optional):

Reviewers' comments:

Reviewer's Responses to Questions

**Comments to the Author**

1. If the authors have adequately addressed your comments raised in a previous round of review and you feel that this manuscript is now acceptable for publication, you may indicate that here to bypass the “Comments to the Author” section, enter your conflict of interest statement in the “Confidential to Editor” section, and submit your "Accept" recommendation.

Reviewer #1: All comments have been addressed

Reviewer #2: All comments have been addressed

2. Is the manuscript technically sound, and do the data support the conclusions?

Reviewer #1: Yes

Reviewer #2: Yes

3. Has the statistical analysis been performed appropriately and rigorously? 

Reviewer #1: Yes

Reviewer #2: Yes

4. Have the authors made all data underlying the findings in their manuscript fully available?

Reviewer #1: Yes

Reviewer #2: No

5. Is the manuscript presented in an intelligible fashion and written in standard English?

Reviewer #1: Yes

Reviewer #2: Yes

6. Review Comments to the Author

Reviewer #1: (No Response)

Reviewer #2: The authors have addressed all of my concerns and I support acceptance of the manuscript at this stage.

7. PLOS authors have the option to publish the peer review history of their article (what does this mean?). If published, this will include your full peer review and any attached files.

Reviewer #1: No

Reviewer #2: No

---

## [Editor Report · Acceptance letter]

1 Sep 2022

PONE-D-22-05923R2 

Associations of Lifetime Concussion History and Repetitive Head Impact Exposure with Resting-State Functional Connectivity in Former Collegiate American Football Players: An NCAA 15-Year Follow-Up Study 

Dear Dr. Walton:

I'm pleased to inform you that your manuscript has been deemed suitable for publication in PLOS ONE. Congratulations! Your manuscript is now with our production department. 

Kind regards, 

on behalf of

Dr. Jacob Resch 

Academic Editor

PLOS ONE